# An adaptive, continuous-learning framework for clinical decision-making from proteome-wide biofluid data

Johannes B. Müller-Reif [1,6,7] ✉, Vincent Albrecht [1,6], Vincenth Brennsteiner[1], Jakob M. Bader [1], Peter V. Treit [1], Nicolai J. Wewer Albrechtsen [2,3], Susanne Pangratz-Führer [4] & Matthias Mann [1,5,7] ✉

Mass spectrometry (MS)-based proteomics provides deep molecular insights from patient samples, but clinical use has been limited by missing values, static biomarker panels, and the need for targeted assay development. We present a framework – Adaptive Diagnostic Architecture for Personalized Testing by Mass Spectrometry (ADAPT-MS) – that enables direct diagnostic and prognostic interpretation of discovery-mode proteomics data at the level of individual samples. ADAPT-MS dynamically retrains simple, robust classifiers based on the proteins quantified in each sample, eliminating the need for imputation or fixed panels. Applied to plasma and cerebrospinal fluid datasets across diseases and clinical centers, it achieves high performance and generalizability using robust, transparent and generalizable statistical models. A single proteomic measurement can support multiple diagnostic questions via retrospective cohort matching, with each classification taking only seconds. As population-scale proteomics datasets grow, this approach lays the foundation for scalable, real-time, and personalized diagnostics directly from proteome-wide data. Such a community effort may help to transform discovery proteomics into a routine clinical tool.

Mass spectrometry (MS)-based proteomics offers unparalleled insights into patient biology, with the ability to quantify thousands of proteins in a single run[1,2]. Plasma proteomics has emerged as a promising approach for understanding systemic disease processes, monitoring health states, and identifying therapeutic opportunities[3–8]. However, despite dramatic improvements in analytical depth, throughput, and robustness, the clinical application of discovery proteomics remains limited[6,9].

The prevailing use of proteomics in biomedicine follows a well-defined biomarker discovery pipeline[10]. First, a sufficiently powered patient cohort is assembled—typically in a case−control or longitudinal design. Next, statistical or machine learning approaches are used to identify proteins differentially abundant between groups. A fixed subset of these proteins is then selected as candidate biomarkers, and targeted assays (e.g., ELISA or targeted MS) are developed and validated in one or more follow-up studies[11,12]. Only after this validation step are biomarker panels applied to new patient samples for diagnostic or prognostic purposes.

Despite its general acceptance, simplicity and wide-spread use, this multistep pipeline has several well-known limitations[9,13,14]. It is

[1]Department of Proteomics and Signal Transduction, Max Planck Institute of Biochemistry, Martinsried, Germany. [2]Department of Clinical Biochemistry, Copenhagen University Hospital - Bispebjerg and Frederiksberg Hospital, Copenhagen, Denmark. [3]Department of Clinical Medicine, Faculty of Health and Medical Sciences, University of Copenhagen, Copenhagen, Denmark. [4]Department of Pediatrics, Dr. von Hauner Children's Hospital, Ludwig-Maximilians-Universität München, Munich, Germany. [5]Novo Nordisk Foundation Center for Protein Research, Faculty of Health Sciences, University of Copenhagen, Copenhagen, Denmark. [6]These authors contributed equally: Johannes B. Müller-Reif, Vincent Albrecht. [7]These authors jointly supervised this work: Johannes B. Müller-Reif, Matthias Mann. ✉e-mail: jomueller@biochem.mpg.de; mmann@biochem.mpg.de

time- and resource-intensive and relies on orthogonal technologies that may not reflect the original discovery data. Therefore, developing a targeted assay for a small panel of proteins requires reagent optimization, calibration, and clinical validation often resulting in long development times[15]. When done by MS, assays are often run on mature triple quadrupole platforms, which, while robust, have not benefited from the recent advances in MS instrumentation[16]. Consequently, rich, multidimensional proteomic data are condensed into a narrow, single-purpose test. Once validated, a given panel addresses only one diagnostic question. If that test returns negative, new samples and new tests are required. While this strategy has produced clinically useful assays, it does not scale well to the complexity of real-world diagnostics, where overlapping syndromes and ambiguous presentations demand flexible, multipurpose tools. ADAPT-MS bypasses this lengthy development cycle entirely by using a single, unbiased measurement universally applicable for diagnostic questions.

A major barrier to using discovery proteomics data directly for diagnostics is the presence of missing values[17,18]. These arise both randomly and non-randomly, typically as a function of protein abundance, sample complexity, and acquisition mode. For example, in data-dependent acquisition (DDA) workflows, low-abundance proteins are more likely to be missed in a semi-stochastic fashion. Standard machine learning pipelines generally assume complete data matrices and struggle with the sparsity typical of large-scale proteomic datasets[19]. Imputation strategies can be applied but often introduce noise or bias that compromises interpretability and generalizability[20]. In contrast, targeted assays typically ensure data completeness by design, enabling even complex classifiers to function reliably. However, by definition they focus on just a few proteins, completely neglecting the remainder of the proteome.

In addition to data 'missingness', discovery proteomics faces other challenges in clinical translation. Quantification is relative, not absolute; quality control and FDR thresholds vary across the proteome; and there is no straightforward way to establish population-level reference ranges for thousands of proteins simultaneously. Moreover, clinical laboratories may lack the infrastructure or expertise to interpret large-scale omics data at the individual level. These issues have so far confined discovery proteomics to a research tool, with clinical applications limited to post-discovery, simplified surrogate assays.

We reasoned that recent developments may open the door to a new paradigm. Advances in sample preparation, instrumentation, and data processing have made high-throughput, robust plasma proteomics feasible at scale[4,21,22]. Large population-based studies are generating proteomic datasets with linked clinical phenotypes and outcomes, establishing a foundation for data-driven diagnostics. At the same time, clinical demand is shifting toward multiplexed, rapid, and hypothesis-flexible tools—an unmet need that proteomics is uniquely suited to address[23,24].

This motivates a fundamental question: Can we build a framework that enables direct diagnostic interpretation of proteome-wide data e.g., from blood plasma, without requiring imputation, fixed biomarker panels, or targeted assays? Such a method would need to handle missingness gracefully, adapt to the proteins actually measured in a given sample, and scale across diseases, populations, and clinical settings.

Here, we present an adaptive diagnostic strategy, termed ADAPT-MS that enables direct classification of individual samples using discovery proteomics data. We evaluate its performance across multiple diseases and clinical centers and benchmark it against conventional approaches. This framework represents a conceptual shift—from fixed, panel-based testing to a dynamic, sample-specific interpretation of the proteome.

Moreover, as underlying proteomic knowledgebases expand, this model will continually refine its diagnostic accuracy—allowing sampling from an increasingly robust range of predictive markers.

## Results

### The ADAPT-MS framework

Current clinical proteomics pipelines rely on a rigid multi-step process: biomarker discovery is performed in a case–control cohort, followed by assay development for a fixed subset of proteins, and only then are these assays applied to classify new patient samples (Fig. 1A). This approach ensures data completeness through targeted measurements but limits flexibility, discards most of the proteome-scale data, and addresses only one diagnostic question at a time.

We developed a framework that enables direct use of discovery proteomics data for diagnostic classification, without the need for targeted assay development or imputation of missing values. The core innovation of our method—termed Adaptive Diagnostic Architecture for Personalized Testing by Mass Spectrometry (ADAPT-MS)—is the dynamic retraining of classification models on a per-sample basis. Rather than applying a fixed classifier across all samples, ADAPT-MS dynamically adapts the model to the proteins actually quantified in each individual sample (Fig. 1B, C). A relaxed feature list is first generated from the discovery dataset using cross-validation and feature selection. For each test sample, the overlap between its measured proteins and this feature list defines a custom feature set, on which a classifier is retrained using the original discovery data. The resulting model is used to assign a diagnostic label or score to that individual case.

This dynamic retraining strategy overcomes the long-standing problem of missing values which are always present in any discovery-mode proteomics. Subsequently, when applying a model on a single sample of interest, we only rely on features (proteins) that are actually detected. In this way, ADAPT-MS sidesteps the intricacies and potential biases of imputation or a simple zero-fill approach. It enables robust single-sample classification without requiring complete data matrices or bias-prone imputation, and allows direct diagnostic use of the proteomics measurement itself on single samples—transforming the output of a discovery experiment into a clinical decision tool.

Importantly, ADAPT-MS does not rely on a pre-defined case-control study for every diagnostic question. Instead, it leverages the increasing availability of large, population-scale proteomic datasets with linked clinical outcomes. For each new diagnostic task, the system can retrospectively assemble a suitable training cohort by identifying samples that match the patient of interest along relevant covariates, such as age, sex, comorbidities, and sampling context. This retrospective "cohort slicing" enables precise, hypothesis-specific modeling without requiring new prospective studies for every differential diagnosis. In effect, the framework replaces the traditional one-question-one-panel paradigm with a flexible, reusable infrastructure that draws diagnostic power from the full depth and scale of previously acquired proteome data (Fig. 1D). Illustrative clinical use cases for this flexible diagnostic approach are summarized in Table 1.

### Dynamic retraining on plasma and serum proteomics cohorts

To evaluate the performance of the ADAPT-MS framework under real-world conditions, we applied it to a recently published dataset of plasma proteomes from patients assigned different subgroups of sepsis and healthy controls. This study, generated by the Roman Fisher group, represents one of the most comprehensive clinical proteomics efforts to date[25]. Among other sub-cohorts, this comprises a discovery cohort (902 samples) and a separate validation cohort (459 samples), all processed using high-throughput DDA in PASEF mode. We focused on a binary classification task: distinguishing sepsis patients with a specific transcriptomically defined sepsis response signature (SRS) from those without[26,27]. This biologically relevant subclassification is clinically challenging and had not been addressed in the original study. The data posed significant technical challenges typical for discovery proteomics, including variable and semi-stochastic missing values, especially for lower-abundance proteins (Supplementary Fig. 1A, B).

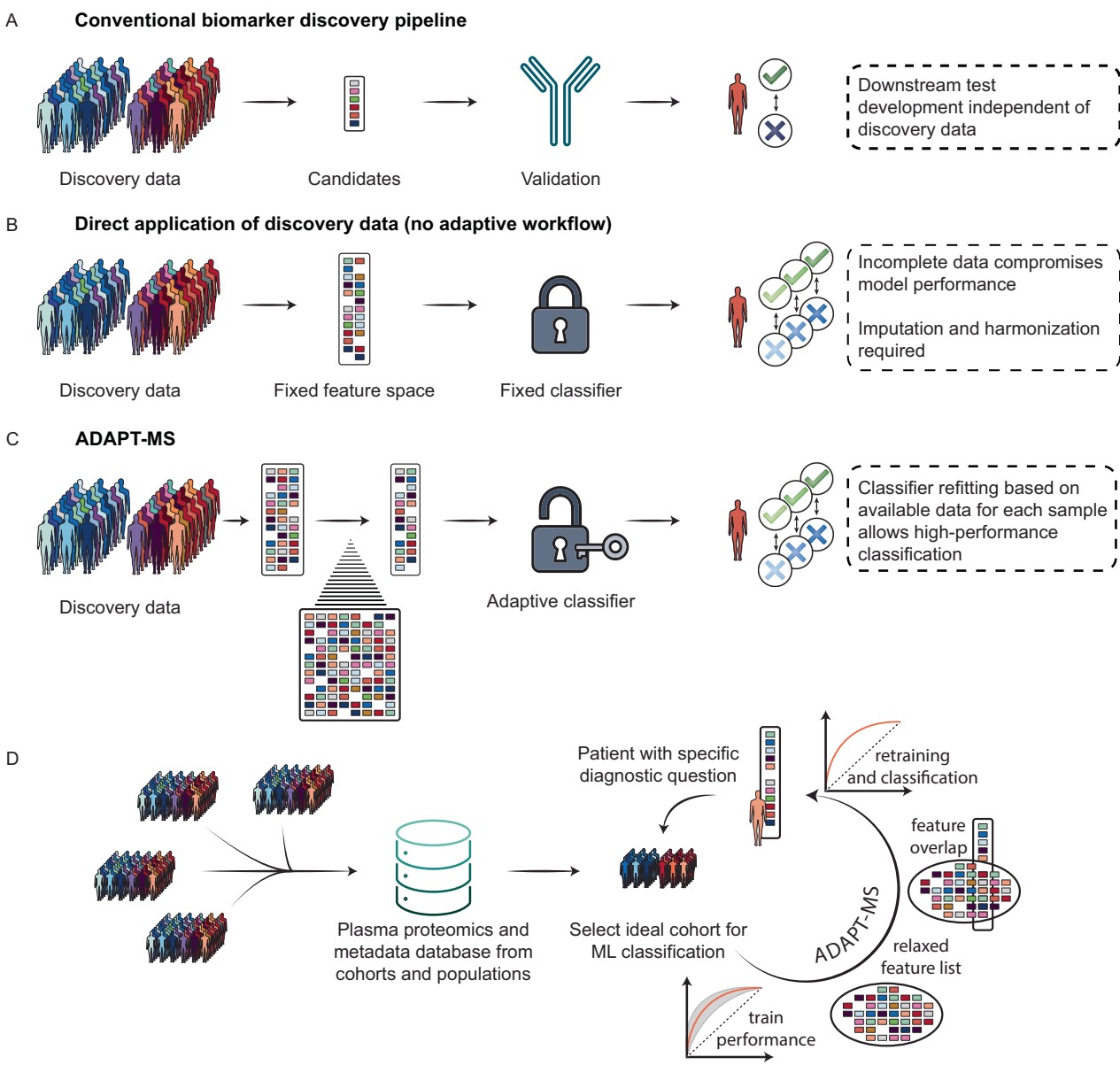

**Fig. 1 | Conceptual overview of ADAPT-MS compared to traditional biomarker pipelines. A** In the conventional proteomics pipeline, candidate biomarkers are identified in a discovery cohort (e.g., case–control study), translated into targeted assays, and applied to new patients for a single diagnostic question. **B** Direct application of discovery proteomics data without adaptation to new patient samples suffers from incomplete data matrices, reducing classifier performance and often requiring imputation or harmonization. **C** ADAPT-MS overcomes this by dynamically retraining the classification model for each patient sample based only on the proteins actually measured, using a relaxed feature set derived from the training data. **D** For new diagnostic questions, ADAPT-MS retrospectively assembles matched training cohorts from large proteomics databases using sample metadata. Feature selection and classifier training are performed on this subset, enabling personalized, scalable, and continuous diagnostic applications based on unbiased proteome-wide measurements.

As a benchmark, we first trained a traditional machine learning model using the full discovery cohort: feature selection by random forest and classification with XGBoost. This pipeline achieved robust performance in cross-validation and yielded an area under the curve (AUC) of 0.83 on the independent validation set, demonstrating that proteomics data can support complex classification tasks when data completeness is ensured.

To apply the ADAPT-MS framework, we first extracted a relaxed set of protein features to support later sample-specific retraining. Relaxed in this context refers to features selected from randomly sampled cohort subsets in a cross-validation-like fashion, introducing controlled variability in the patient population and corresponding feature sets. This variability increases robustness in later patient-specific selection of features—by reducing dependence on any specific cohort composition. We performed a grid search over feature selection methods and classifiers (Supplementary Fig. 1C, D). As performance was consistent across most combinations (AUC 0.80–0.83), we selected t-test-based feature selection combined with logistic regression for its simplicity, robustness, and interpretability[28]. However, the ADAPT-MS framework is agnostic to both the feature selection method and classification algorithm employed. Classification tasks requiring non-linear decision-boundaries can readily incorporate algorithms, such as random forest or gradient boosting within the same ADAPT-MS architecture, simply by substituting the classifier while maintaining the sample-specific retraining strategy.

**Table 1 | Illustrative clinical use cases for ADAPT-MS**

| Scenario | Presentation | ADAPT-MS Application | Advantages over status quo |
|---|---|---|---|
| **Specific diagnostic test** | Symptoms or condition and rule-in/rule-out diagnosis, requiring differential diagnosis or assessment of severity | Single proteome measurement and classification based on proteome database | Any condition that is sufficiently represented in the database can be probed robustly. Tests can be adjusted to local/individual specific or selection criteria |
| Example 1: Sepsis Triage in Emergency Departments | Acute presentation with suspected infection; unclear source and severity | Single plasma proteome measurement used for sepsis vs non-sepsis classification or sepsis subtyping | Faster diagnosis, risk stratification, targeted treatment initiation, potential improvement in survival |
| Example 2: Early Detection of Alzheimer's Disease in Memory Clinics | Mild cognitive impairment with inconclusive standard biomarker results | CSF proteome profiling with dynamic cohort matching for Alzheimer's classification | Higher diagnostic certainty, earlier intervention with disease-modifying therapies |
| **Non-specific diagnostic test (screening)** | Clinical presentation does not distinguish at-risk individuals in a general population (e.g., fetal growth restriction in first trimester as stratification for small for gestational age condition at birth) | Single plasma proteome measurement and assessment by classifier based on database | Any condition that is sufficiently represented in the database can be probed robustly. Tests can be adjusted for high negative predictive or high positive predictive metrics as needed. |
| Example: Monitoring Multimorbidity in Chronic Disease Management | Chronic conditions, such as diabetes, obesity, and kidney disease requiring regular follow-up | Periodic plasma proteome analysis to detect early signs of comorbid progression or new risks (e.g., cardiovascular events) | Personalized care adjustments, preemptive interventions, reduction in hospitalizations |

For each of the individual samples, we then identified the intersection between this relaxed feature list and the proteins actually quantified. A logistic regression model was then retrained on the discovery data using this intersection and used to classify the individual sample. This procedure was repeated independently for all validation samples. The ADAPT-MS model slightly outperformed the conventional fixed-classifier approach, reaching an AUC of 0.87 (Fig. 2A). Importantly, the performance remained stable even in samples with relatively high missingness, without requiring imputation. We further explored the number of features contributing to each prediction and found that performance plateaued at around 50 proteins per sample (Supplementary Fig. 1E). Correctly and incorrectly classified samples drew from similar distributions of available features over a broad range of requested features from feature selection, highlighting the method's robustness to varying data completeness (Fig. 2B, Supplementary Fig. 1F). To further confirm the robustness of ADAPT-MS, we applied its retraining pipeline across different training set sizes and found that it consistently achieved high performance, comparable to conventional XGBoost classifiers on the same data (Fig. 2C). Computational time increased with training set size for both methods (Supplementary Fig. 1G, H). ADAPT-MS showed somewhat steeper growth during training, reflecting repeated feature-selection steps, but remained computationally lightweight overall (a one-time computational cost that takes only minutes on a standard laptop) and was faster than XGBoost during validation.

To illustrate the extension of ADAPT-MS to prognostic classification tasks, we applied our architecture to a recently published cohort investigating the risk prediction for development of metabolic syndrome (MetS) from serum proteomics[29]. Both discovery and validation cohorts were measured using unbiased discovery proteomics, making this dataset an ideal example case. The authors of the study applied a complex ML architecture wrapped around LightGBM gradient boosting; thus, it is not surprising that a standard RF−XGBoost model does not perform on par (Fig. 2D). However, the ADAPT-MS architecture yields equivalent performance to the published model (AUC 0.77) on the validation data, demonstrating the power of our simple and explainable yet readily applicable model (Fig. 2E).

These results demonstrate that ADAPT-MS can match or exceed standard machine learning pipelines while enabling true single-sample diagnostics and prognostics directly from discovery-mode proteomics. Our dynamic retraining strategy ensures that each classification is optimized for the data available in a given patient sample,

making it suitable for routine clinical use even in the presence of variable missingness.

## Generalizability of ADAPT-MS across clinical centers in CSF proteomics

To evaluate the robustness and cross-site generalizability of ADAPT-MS, we applied it to a multicenter cerebrospinal fluid (CSF) proteomics dataset of Alzheimer's disease (AD) patients and controls. This dataset, generated in our own laboratory, comprises three independent clinical cohorts from Sweden, Berlin, and Magdeburg/Kiel, processed under comparable protocols but representing real-world variability in patient populations, pre-analytical bias and diagnostic accuracy[30]. The variability is visible by cohort-specific proteome signatures in unsupervised methods (Supplementary Fig. 2J) as well as in the different numbers of significantly changing proteins between AD patients and control samples across the different cohorts.

Each cohort included CSF proteomes acquired by data-independent acquisition (DIA) and annotated with standard AD biomarkers and clinical diagnoses. As previously reported, the Sweden and Magdeburg/Kiel cohorts showed clear separation between AD and non-AD patients, whereas the Berlin cohort exhibited substantial overlap, reflecting the challenge of cross-center reproducibility in clinical proteomics.

We first reanalyzed the raw data using an updated library-free DIA pipeline and reproduced the original classification performance using conventional machine learning (mean AUC 0.96, Supplementary Fig. 2A). Building on this, we applied ADAPT-MS using the Sweden cohort for discovery, with relaxed feature selection via cross-validation and t-tests (performance on 5x CV training cohort Sweden AUC 0.96, Supplementary Fig. 2B). We then dynamically retrained a logistic regression classifier for each individual test sample in the Berlin and Magdeburg/Kiel cohorts based on its quantified proteins and compared it to full dataset classification based on XGBoost following data frame imputation. ADAPT-MS achieved strong generalization performance across both external sites. When training on Sweden and testing on the other two cohorts, the AUC reached 0.85 (Magdeburg/Kiel) and 0.73 (Berlin)−despite the lower diagnostic signal in the latter cohort (Fig. 3A Supplementary Fig. 2C, D). Performance improved when training on a combined discovery cohort of Sweden and Magdeburg/Kiel, reaching an AUC of 0.80 in Berlin (Fig. 3A, Supplementary Fig. 2E, F). This demonstrates that ADAPT-MS benefits from diverse training data and scales well to complex, real-world settings. We also tested a multicentric diagnostic setup, where the training cohort was

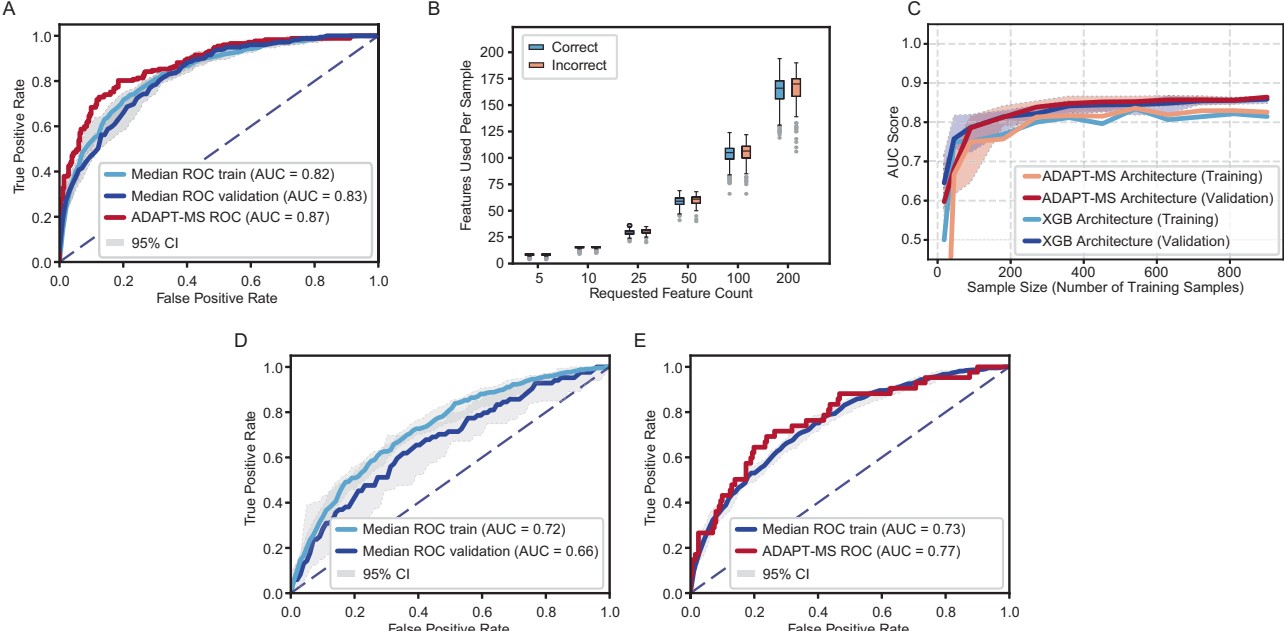

**Fig. 2 | Application of ADAPT-MS to plasma and serum proteomics cohorts for diagnostic and prognostic classification. A** Receiver Operating characteristic (ROC) analysis comparing ADAPT-MS to conventional fixed feature pipeline: ADAPT-MS outperforms a conventional fixed-feature pipeline (AUC 0.87 vs. 0.83). Shaded areas represent 95% confidence intervals from 10x repetition of classification for train and validation sets. The ADAPT-MS classifier is deterministic and has no defined confidence interval. **B** Robustness to missing values: correctly and incorrectly classified samples have similar numbers of features used, indicating consistent performance across varying degrees of missingness. Boxplots show median (line), interquartile range (box), and 1.5x interquartile range (whiskers) for the number of features used for classification per sample (461 samples classified in total in from the validation set). **C** Learning curves showing area under the ROC curve (AUC) as a function of training sample size for ADAPT-MS (Refit Architecture) and conventional XGBoost (XGB). Classification performance improves with increasing training set size for ADAPT-MS, highlighting its ability to benefit from continuous data accumulation. Shaded areas show 95% confidence interval from ten iterations. **D** RF – XGBoost architecture performance for the training and validation cohort sets of the MetS study. Gray areas show 95% confidence interval from ten iterations. **E** ADAPT-MS performance on the training cohort for the training architecture part and in single sample-retraining mode for the validation cohort. Gray areas show 95% confidence interval from ten iterations for the training data.

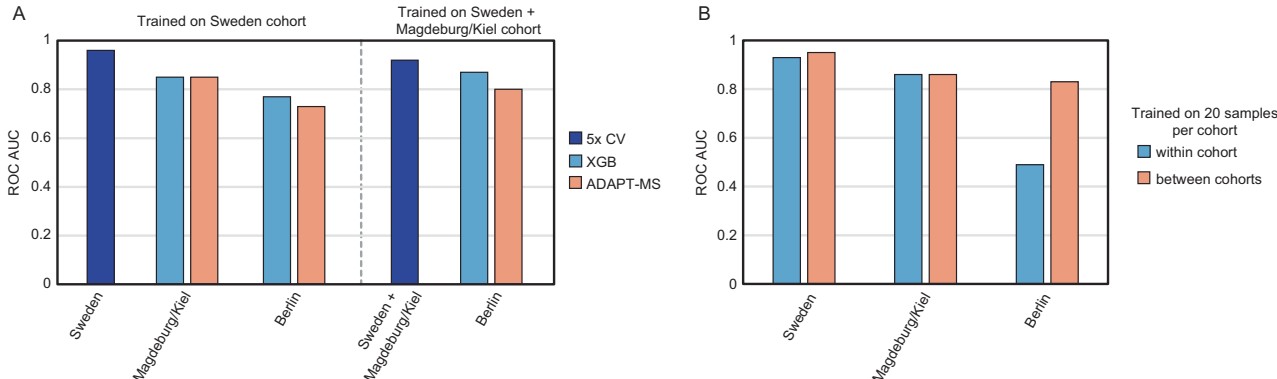

**Fig. 3 | ADAPT-MS enables robust cross-center classification in a multicenter CSF proteomics study of Alzheimer's disease. A** Classification performance (ROC AUC) of ADAPT-MS compared to fixed-feature and XGBoost classifiers in two cross-center settings: training on Sweden alone, and training on Sweden combined with Magdeburg/Kiel, both tested on Berlin. ADAPT-MS outperforms or matches alternatives across scenarios. **B** Performance of ADAPT-MS using only 20 training samples per cohort, comparing within-cohort and cross-cohort classification. ADAPT-MS achieves consistently high AUCs even in cross-site settings, highlighting its adaptability across heterogeneous datasets without the need for imputation or panel harmonization.

constructed from a balanced subset of patients from all three clinical sites. This strategy yielded strong and stable classification across cohorts, highlighting the flexibility of the framework to integrate heterogeneous data sources. Importantly, the method does not rely on full-cohort imputation or harmonization for any validation or actual application case on single samples, but adapts per sample based on the available feature space (Fig. 3B).

Together, these results underscore a key advantage of ADAPT-MS: the ability to support individualized diagnostics across clinical sites,

even in the presence of technical and biological variation. This cross-center generalizability is essential for real-world deployment of proteomics-based diagnostics.

## Retrospective cohort matching improves classification performance in simulated diagnostics

To further evaluate the retrospective cohort selection component of the ADAPT-MS framework, we again turned to the sepsis proteomics dataset used above. While the SRS signature has been defined by

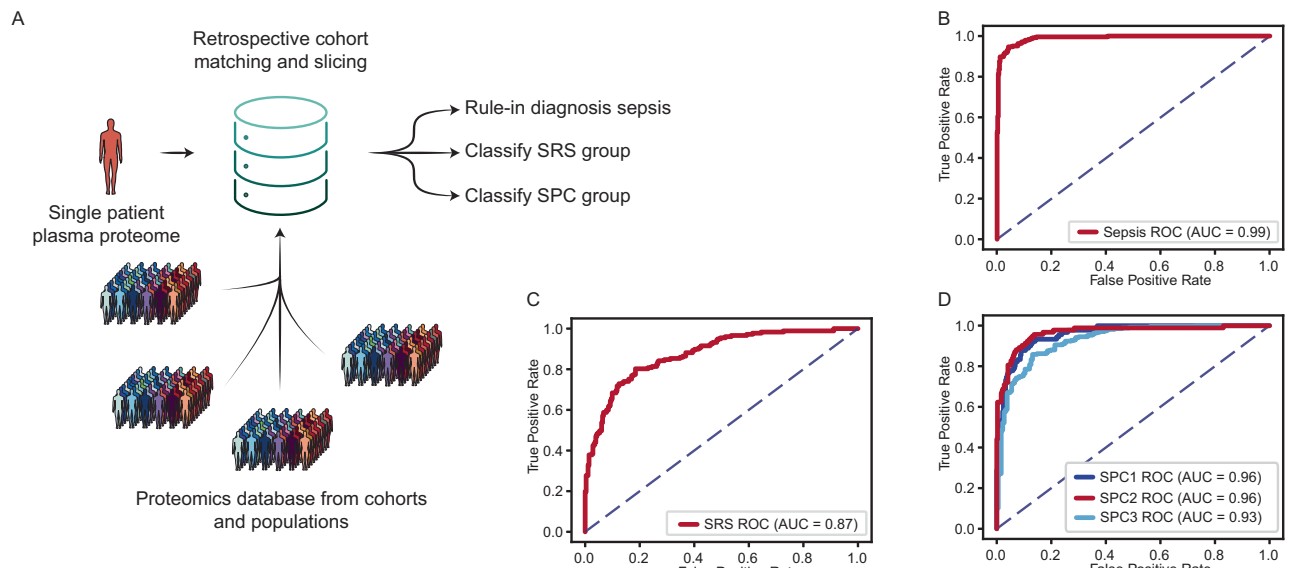

**Fig. 4 | Retrospective cohort matching enables flexible, personalized, and accurate diagnostics from proteome-wide data. A** Schematic of the ADAPT-MS workflow for retrospective cohort matching. A single patient's proteome-wide measurement is reused to address multiple diagnostic questions. For each task, a matched training cohort is dynamically selected from a reference database using metadata (e.g., age, sex, comorbidities), and a sample-specific classifier is retrained. **B** Classification performance of ADAPT-MS for sepsis vs controls (AUC = 0.99). **C** Classification performance of ADAPT-MS for SRS vs non-SRS (AUC 0.87). **D** Classification performance of ADAPT-MS for sepsis plasma proteome-based clusters (SPC1/2/3), with AUCs ranging from 0.93 to 0.96.

transcriptomics to classify patients with different forms of sepsis and proteomics is able to predict those classes well (Fig. 2), Mi et al. defined a purely proteome-based classification of sepsis patients based on unsupervised proteome clustering (SPC1/2/3)[25]. This illustrates the advantage of ADAPT-MS, being able to adaptively and retrospectively *slice and match cohorts* for diagnostic decisions. Based on the proteomics data, any newly acquired patient sample can be classified into i) sepsis or non-sepsis, based on a classifier built on the cases vs control samples of the cohort/database, ii) SRS or non-SRS type of sepsis and iii) into the newly defined proteomics sepsis classes (SPC1/2/3) from Mi et al. (Fig. 4A–C). Further, any other covariate present with sufficient biological effect size could be added to the possible classifications. For example, if the control samples would include a large number of patients with acute inflammation but not sepsis, this would be possible to differentiate and diagnose with a separate classifier.

This highlights a key strength of ADAPT-MS: tailoring diagnostics to individual cases using retrospective matching. In real-world scenarios where diseases are heterogeneous and prospective control cohorts are impractical, this approach would enable flexible, scalable diagnostics grounded in existing data. It also addresses the n × m scaling problem in diagnostic development by reusing the same sample against dynamically assembled reference groups—supporting multiple hypotheses without remeasurement.

Crucially, these gains come with no increase in computational or practical complexity. Each retraining step takes only a few seconds, even on standard hardware, and uses simple, well-established statistical methods, such as ANOVA for feature selection and logistic regression for classification. This deliberate simplicity promotes interpretability and generalizability, avoiding the overfitting often associated with complex, highly parameterized machine learning or deep-learning models. By design, the system is transparent and stable, favoring robustness over marginal improvements in performance.

The flexibility of retrospective cohort matching further enhances diagnostic resolution. Depending on the clinical context, the reference population can be broadly defined—e.g., healthy individuals of the same age and sex—or finely tuned to match the characteristics of the patient under consideration. This includes the possibility of constructing a near-identical comparator group, effectively creating a "digital twin" for highly personalized diagnostics. Because the full proteomic profile is measured once and retained, each sample can be interpreted repeatedly and also longitudinally, against different reference groups and for multiple diagnostic hypotheses, without remeasurement or assay development.

## Discussion

We present ADAPT-MS, a conceptual framework that enables diagnostic and prognostic classification directly from discovery-mode proteomics data at the level of individual patient samples. By dynamically retraining the classifier on the proteins actually detected in each sample, the method avoids imputation, fixed biomarker panels, and targeted assay development. Across two clinical contexts—plasma proteomics in sepsis and CSF proteomics in AD—we demonstrate that ADAPT-MS performs on par with or better than conventional machine learning approaches, while offering far greater flexibility and scalability. Importantly, the framework supports robust performance across clinical centers and heterogeneous cohorts, addressing a long-standing challenge in the clinical application of omics technologies.

Importantly, we do not claim that ADAPT-MS outperforms other ML techniques in their typical applications, such as analysis of complete datasets in rearch settings. Rather, its novelty lies in enabling single-sample diagnostic application of discovery proteomics data—a use case for which no other framework currently exists.

This work proposes a shift from the prevailing paradigm of biomarker discovery, where fixed panels are developed through long validation pipelines, to a model where the full proteome is interpreted directly and flexibly for diverse diagnostic questions. Instead of discarding the majority of the measurement after marker selection, ADAPT-MS allows clinicians to extract multiple inferences from the same data, using dynamically retrained, sample-specific models. This mirrors the complexity of real-world clinical decision-making, where differential diagnoses often evolve over time.

We believe the key advantage of ADAPT-MS is not to replace targeted assays for established biomarkers—which remain the gold standard for specific diagnostic applications—but to enable

capabilities that targeted approaches cannot. First, unlike targeted assays that answer a single clinical question, ADAPT-MS can support multiple differential diagnoses from one measurement, which is particularly valuable when clinical presentations are complex or ambiguous. Second, it allows rapid deployment for emerging diagnostic needs (e.g., novel disease subtypes or pathogens) without the lengthy development cycle required for targeted panels. Third, as reference databases expand, ADAPT-MS applications improve continuously, whereas targeted assays remain fixed once validated.

Technically, ADAPT-MS introduces a strategy for handling missing data without imputation. The core innovation—sample-specific model retraining—improves robustness, and benefits from growing reference databases. Once a high-quality proteome is acquired, classification tasks can be executed without new assays, using retrospective cohort matching and tailored model retraining. This reuse of data rather than redesign of tests could, in principle, streamline diagnostics and accelerate response times in complex clinical scenarios. ADAPT-MS benefits from increasing data availability: as more samples with outcome labels are added to the reference pool, future predictions improve, creating a self-reinforcing diagnostic system.

In complex cases, such as neonatal sepsis, ICU triage, or multimorbidity in elderly patients, the ability to generate multiple differential diagnoses from a single proteomic measurement could substantially reduce time to treatment and improve clinical decision-making. ADAPT-MS also addresses ethical challenges that arise with exploratory omics: because only clinician-requested models are applied to the data, it avoids unsolicited or non-actionable findings, thus aligning broad molecular measurement with focused clinical intent[31].

However, ADAPT-MS is currently mainly a conceptual advance rather than an already deployed clinical tool. Several challenges must be addressed to move from proof-of-principle to real-world impact. For example, the utility of retrospective cohort matching depends heavily on access to large, well-annotated proteomic datasets with standardized sample processing and reliable outcome data, resources that remain limited in scope and quality. Perhaps more critically, the adaptive nature of the model raises regulatory and operational complexities. Diagnostic frameworks that evolve over time or differ per patient challenge current validation and approval processes, and require new thinking around performance auditing, version control, and clinical interpretability.

Some of those challenges are intrinsically handled by ADAPT-MS, including the ethical concerns inherent in the analysis of comprehensive omics data[32]. An example are the incidental findings—discovering conditions that were not part of the original diagnostic query, such as the discovery of a BRCA1 mutation without informed consent[33]. ADAPT-MS effectively circumvents this issue: although the entire proteome is measured, only specific diagnostic models requested by clinicians are applied to the data. This prevents algorithmic exploration that might generate unintended or unconsented findings, while still leveraging the benefits of data-rich discovery proteomics. Clinicians receive only classification results relevant to their specific diagnostic hypothesis, maintaining ethical boundaries with a comprehensive measurement framework.

From a regulatory perspective, ADAPT-MS may offer a more direct path to clinical implementation than previous proteomics approaches. It maintains the integrity of the original measurements without requiring imputation or data harmonization. Additionally, by facilitating algorithmic use of well-established and robust statistical and machine learning methods like t-test and logistic regression, ADAPT-MS provides interpretability and transparency that complex "black box" approaches lack.

Nevertheless, important regulatory considerations remain. Clinical implementation will require robust validation studies demonstrating reproducibility, accuracy, and generalizability across diverse healthcare settings. The adaptive nature of our algorithm, while providing performance advantages, necessitates thoughtful approaches to test "locking" and version control that satisfy regulatory requirements while preserving the benefits of iterative improvement. Addressing these challenges will require collaboration with regulatory authorities, clinical laboratories and industrial partners. The transition from academic exploration to clinical implementation will remain a challenge, but the potential benefits—improved diagnoses, faster time-to-result and time-to-treatment, and efficient use of healthcare resources—make this an area worth further exploration. ADAPT-MS offers a promising approach for bringing discovery proteomics closer to routine clinical practice. We therefore envision ADAPT-MS as a starting point for iterative development and cautious clinical exploration. Initial applications may be most appropriate in research settings or for unmet diagnostic needs where no effective assays currently exist. Wider adoption will depend not only on technological maturity but also on the evolution of clinical, infrastructural, and regulatory ecosystems.

To advance ADAPT-MS from a conceptual framework to a clinically validated diagnostic tool, a phased regulatory strategy aligned with, e.g., the European In Vitro Diagnostic Regulation (IVDR) is essential. Two primary implementation pathways are available under IVDR: (i) deployment as an in-house diagnostic device—commonly referred to as a laboratory-developed test (LDT)—manufactured and used solely within health institutions under Article 5 or (ii) certification as a CE-marked in vitro diagnostic (IVD) device through a notified body. While the former enables early clinical use and fosters innovation, it comes with stringent requirements, including ISO 15189 compliance, appropriate quality management systems, and the demonstration that no equivalent CE-marked device meets the specific needs of the patient population. These routes provide structured pathways to evaluate clinical validity, reproducibility, and integration into diagnostic workflows. In parallel, prospective studies targeting high-impact use cases, such as sepsis triage or early Alzheimer's diagnosis can help establish clinical utility. We envision that laboratory-developed tests would be implemented first, followed by more general approval, as laboratory approved tests are increasingly disfavored in the major regulatory regions. Importantly, the transparent, interpretable architecture of ADAPT-MS aligns well with growing regulatory expectations for explainability and auditability in clinical AI tools[34–36].

When competing with established diagnostic procedures, classical diagnostic metrics, such as negative predictive value, positive predictive value, sensitivity and specificity are the performance characteristics by which clinical success will be measured[37]. With ADAPT-MS the clinician can select the desirable metrics according to the case at hand. For instance, a test can be specifically selected and trained e.g., for rule-out criteria with a high negative predictive value as main priority. We propose that in future, diagnostic test will be specifically ordered with statistical preference tailored to individual patients' needs[38], governed by the nature of the analytical question that can be either diagnostic or prognostic and either screening or informed diagnostic testing. In practice, the lines between these categories are often blurred and ADAPT-MS can flexibly target tests to specific use cases. However, it is clear that for implementation in clinical practice, this principle will have to be applied to real-world validation studies before our presented vision of the tailored single sample use-case can be approved and readily applied.

Looking ahead, the potential of this approach is considerable. As longitudinal, high-throughput proteomics becomes feasible at population scale and decreasing cost, individual-level proteome profiles may become part of routine clinical records. In that context, ADAPT-MS could enable real-time diagnostic interpretation by matching each new patient to precisely defined reference cohorts and applying flexible, task-specific models. This shift from assay-centric to data-centric

diagnostics would align proteomics more closely with the model already established in clinical genomics and could support a new level of precision in medical decision-making.

In summary, ADAPT-MS introduces a conceptual framework for making diagnostic use of proteome-wide data without the constraints of fixed panels or assay development. While significant hurdles remain, especially for clinical implementation, the combination of sample-specific retraining and retrospective comparator matching offers a promising path toward flexible, data-driven proteomic diagnostics.

# Methods

## Data processing

All proteomics datasets were obtained from publicly available repositories a described in the Data Availability section.

## Sepsis plasma proteomics analysis

**Data preprocessing.** Protein intensities were log10-transformed. Proteins missing in more than 5% of samples were removed to ensure basic data coverage. Where complete data matrices were required, remaining missing values were imputed using k-nearest neighbor (KNN) imputation ($k = 5$, Euclidean distance).

**Machine learning approaches.** We applied three classification strategies:

- Baseline fixed-feature models: imputed datasets were used to train classifiers with feature selection performed via random forest models, followed by classification using XGBoost. This conventional approach assumes complete feature matrices across samples.
- ADAPT-MS dynamic retraining: in the ADAPT-MS framework, a relaxed feature list was first derived via univariate t-tests corrected for multiple testing. For each individual validation sample, a logistic regression model was retrained de novo using only the proteins actually quantified in that sample. No imputation was applied at validation time, allowing true sample-specific adaptation.
- ADAPT-MS multiclass classification: ADAPT-MS multiclass feature selection was based on pairwise t-tests across all class combinations, ensuring robust discrimination across multiple sepsis-related classes. Multiclass models were built using one-vs-rest logistic regression for each individual sample and retrained de novo using only the proteins actually quantified in that sample.

Model performances were evaluated using stratified 5-fold cross-validation within the training cohort, and separately on an independent validation cohort. Validation based on ADAPT-MS architecture yielded deterministic classifiers and therefore no error or confidence interval is determined.

## MetS plasma proteomics analysis

**Data preprocessing.** Protein intensities were log10-transformed prior to analysis. To ensure comparability across samples, proteins missing in more than 5% of samples were excluded. Where complete data matrices were required, missing values were imputed using KNN imputation ($k = 5$, Euclidean distance). For validation in the ADAPT-MS framework, no imputation was applied at test time; instead, only the proteins actually measured in a given sample were used for model retraining.

**Machine learning approaches.** Baseline Fixed-Feature Model and ADAPT-MS dynamic retraining were applied as stated above for the sepsis cohort.

Performance was assessed using stratified 5-fold cross-validation within the discovery cohort, and separately on independent validation cohorts. For baseline models, aggregated ROC curves with confidence intervals were derived from repeated cross-validation and external validation. ADAPT-MS retraining yielded deterministic classifiers at the single-sample level, producing sample-specific ROC analyses without imputation.

## AD CSF proteomics analysis

**Data preprocessing.** MS-raw data obtained from the PRIDE database were re-analyzed using DIA-NN 1.8.1 with an in silico predicted library on a high-performance computing cluster searching against the human SwissProt FASTA database (taxonomy ID 9606) downloaded from UniProt in June 2023. All settings were maintained at their default values[39].

Analysis was done in three different modes: For complete cohort analysis all MS-raw files were processed together. For ADAPT-MS and comparative analyses of sets of samples as validation type, the discovery cohort part was processed initially and all validation samples processed with the resulting spectral library from the corresponding discovery set (for example Sweden cohort as discovery set and Magdeburg/Kiel + Berlin cohort as validation set). For subsequent classification tasks either the complete validation protein group matrix was used for traditional ML tasks or the first search protein group tables per single sample were used in the ADAPT-MS architecture.

Protein intensities were log10-transformed for analysis and classification. Where complete data matrices were required, missing values were imputed using KNN imputation ($k = 5$, Euclidean distance). For sample-specific retraining approaches, no imputation was performed for test samples; models were dynamically retrained on the available features.

**Machine Learning Approaches.**

- Baseline fixed-feature models: random forest classifiers were used for feature preselection, followed by model training with XGBoost. Models assumed a complete imputed dataset and fixed feature sets across training and validation.
- ADAPT-MS dynamic retraining: similar to the sepsis analysis, the ADAPT-MS framework was applied by retraining a logistic regression model individually for each test sample, using only the measured subset of proteins. Relaxed feature sets were determined using t-tests on the training data.

**Cross-cohort generalization.** To assess real-world applicability, models were trained on the Sweden cohort and tested on Berlin and Magdeburg/Kiel cohorts. In these settings, two approaches were compared:

- NonRefitClassifier: features were selected in the training cohort and applied without model retraining.
- ADAPT-MS Retraining: models were dynamically retrained per sample based on the overlap between selected features and the measured proteins in each test case.

Model evaluation included stratified 5-fold cross-validation within training sets and ROC-AUC calculations on external cohorts, with confidence intervals derived from repeated resampling where applicable.

**Computational environment.** All analyses were performed in Python (version 3.8) using standard scientific computing libraries including scikit-learn, xgboost, numpy, pandas, scipy, and matplotlib. ADAPT-MS classifiers to automate dynamic feature selection, model retraining, and performance evaluation workflows were implemented on architecture level.

## Reporting summary

Further information on research design is available in the Nature Portfolio Reporting Summary linked to this article.

## Data availability

All proteomics datasets were obtained from publicly available repositories. Plasma proteomics data for the sepsis cohort were downloaded from the PRIDE repository under accession number PXD039875. Serum proteomics data for the MetS cohort were downloaded from the authors resource page (https://guomics.com/publications/2023crm_caix/) and proteomics raw data is available at the PRIDE repository under identifiers PXD039236, PXD039231 and PXD038253. CSF proteomics data for AD were downloaded from PRIDE under accession number PXD016278. Source data are provided with this paper.

## Code availability

The ADAPT-MS ML functions and analysis notebooks to reproduce the publication figures is supplied as GitHub repository: https://github.com/johannesbmueller/ADAPT-MS[40].

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

## Acknowledgements
We thank all members of the Department of Proteomics and Signal Transduction for help and discussions and particular Magnus Schwörer for guidance regarding the software setup. We further thank all research groups providing publicly available proteomics and metadata of patient cohorts.

## Author contributions
Conceptualization: J.M.R. and M.M. Methodology: J.M.R., V.A., J.B., N.W.A., S.P.F., and P.V.T. Software: J.M.R. and V.B. Formal Analysis: J.M.R. and V.A. Writing – Original Draft: J.M.R., V.A., and M.M. Writing – Review & Editing: J.M.R., V.A., M.M., J.B., P.V.T., V.B., N.W.A., and S.P.F. Visualization: J.M.R. All authors read and approved the final manuscript.

## Funding

## Competing interests
MM is an indirect shareholder of Evosep. All other authors declare no competing interests.
