## [Transparent Peer Review file · Nature Communications]

An adaptive, continuous-learning framework for clinical decision-making from proteome-wide biofluid data

Corresponding Author: Professor Matthias Mann

Version 0:

Reviewer comments:

Reviewer #1

(Remarks to the Author)

The manuscript proposes the ADAPT-MS framework for sample-specific model retraining on discovery-mode proteomics data. While the concept of avoiding fixed biomarker panels is innovative, a number of methodological choices and presentation issues must be addressed before the work is suitable for publication.

Major Comments

1. In the first “fixed feature selection” stage the authors carry out full-cohort imputation, yet this imputed data strongly influences the downstream relaxed feature list. What evidence supports applying imputation only at this stage and not during the subsequent single-sample retraining? A detailed justification—including sensitivity analyses that vary the initial imputation strategy—would strengthen the manuscript.
2. Instead of dropping unobserved proteins from the fixed feature list and retraining, why not retain the original fixed model (e.g., the XGBoost model presented in the text) and simply set absent feature values to zero? Please report the diagnostic performance of this “zero-fill” approach so that readers can judge the added value of sample-specific retraining.
3. The authors cite algorithmic simplicity as the reason for choosing logistic regression, yet logistic regression is a linear classifier. What empirical or theoretical evidence suggests that linear decision boundaries are adequate for these proteome-wide clinical tasks? A comparison to at least one nonlinear model under identical training conditions is warranted.
4. The discussion highlights the ability to reuse discovery-mode data for exploratory diagnostics, but does not convincingly explain why this is preferable to more economical targeted proteomics once a clinically actionable protein set has been identified. Please clarify the cost-benefit rationale.
5. The Abstract states that the framework enables both diagnostic and prognostic interpretation, yet the manuscript offers no explicit prognostic example. Please provide at least one prognostic use-case (e.g., survival or treatment response prediction) or temper the claim.
6. In Fig. 2C the training-set performance of ADAPT-MS decreases as the number of training samples increases. This counter-intuitive trend needs explanation—does it reflect over-fitting at small sample sizes, class-imbalance effects, or something else?
7. Line 249 claims that the method “does not rely on full-cohort imputation or harmonization,” yet the relaxed feature selection explicitly uses full-cohort imputation. This sentence should be revised to accurately describe the workflow.
8. ADAPT-MS performs worse within the Berlin cohort than between cohorts. Please explain this pattern (e.g., batch effects, cohort heterogeneity) and include the corresponding fixed-feature XGBoost results for parity.

Minor Comments

1. Line 194: “Extended Data Fig. 2E” should read “Extended Data Fig. 1E.”
2. Clearly state the training and test sample counts for the Alzheimer’s disease cohorts.
3. Lines 271–273: Add the full reference for Mi et al.

(Remarks on code availability)

Reviewer #2

(Remarks to the Author)

(Remarks on code availability)

Reviewer #3

(Remarks to the Author)

Improved MS instrumentation has markedly increased the depth of proteomic information that can be obtained from a single sample and enables high-throughput workflows, building up on population databases. Additionally, the potential of machine learning models for the analysis of complex omics data is omnipresent. At present, many studies report on the use of machine learning models for clinical decision-making based on proteomics data.

In this manuscript, the authors present a framework for adaptive model generation. In principle, a classification model is first trained on reference data using an initial list of proteins. Before the model is applied to a single, unlabeled test sample, this list of proteins is adjusted if these proteins are missing in the test sample. The adjusted list is then used to re-train the model on the reference data before classification of the test sample.

In a first example of application, the authors demonstrate the classification of sepsis patients based on plasma proteomics data. Conventional classification models using the XGBoost algorithm and ADAPT-MS models both show good performance with AUC values in the range of 0.83-0.87. As shown in Fig. 2C, the performances of XGBoost and ADAPT-MS models slightly vary depending on the number of samples in the training set. XGBoost models perform better for training sets of 200-500 samples and ADAPT-MS performs better for training sets of 500-900 samples, though the difference in performance is small.

In a second example, the authors claim to showcase generalizability across three clinical centers that represent real-world variability. For the application of machine learning models in clinical proteomics, the integration of datasets from independent sites is indeed a crucial challenge. However, the data provided in Fig. 3A shows that XGBoost classification models without adaptive learning match or perform better than ADAPT-MS models in all cases. In addition, page 8, line 224 states that the independent cohorts were processed under “comparable protocols”. While this might be true in terms of sample collection at the different clinical sites, all samples were processed in the same laboratory using identical protocols. In the original publication by Bader et. al, the Methods sections provides information that all cohort samples were processed together and MS data was acquired on the same instrument using the same settings. Yet, most variability in proteomic data results from different sample preparation protocols and different MS instrumentation. In my opinion, the data presented in the manuscript therefore does not demonstrate “real-world variability” in proteomics data of independent sites. The application of a classification model, which was trained on a reference dataset, and tested on a single, independent sample of different origin, is still questionable. I would encourage the authors to demonstrate the application of the developed software on datasets that were actually processed independently in different clinical sites.

In summary, the authors provide a concept on how to enable fast use of proteomics discovery cohorts without the need for time-consuming analysis of validation cohorts. As the authors describe in the Discussion, implementation for clinical decision-making requires robust and reproducible validation. However, adaptive learning is quite the opposite and will hardly meet regulatory requirements. Further, the data provided in this manuscript does not sufficiently demonstrate the advantage of adaptive classification models over conventional, previously published approaches. Especially in the context of clinical decision-making that requires strict, yet important regulations. Further, the strategy of adjusting feature lists and re-training models provides little novelty compared to ever-advancing machine learning approaches in proteomics data analysis. In addition, the comparison of computation expense to existing algorithms has not been shown.

The concept of the presented classification algorithm is intriguing and the algorithm certainly deserves publication. However, given the aforementioned constraints, it might be adequate to rather focus on the bioinformatic advancement rather than a very speculative clinical application. A more specialized journal might be a more adequate platform.

Further notes:

- Author affiliation 5 is not mentioned for an author
- p. 7, l. 194 “Extended Data Fig. 2E” must be replaced with “Extended Data Fig. 1E”

(Remarks on code availability)

Reviewer #4

(Remarks to the Author)

(Remarks on code availability)

Version 1:

Reviewer comments:

Reviewer #1

(Remarks to the Author)

With all previous issues having been satisfactorily addressed by the authors, the manuscript is recommended for acceptance. One minor correction is requested - the upper panel of Figure 3 should be removed.

(Remarks on code availability)

Reviewer #3

(Remarks to the Author)

The authors have mostly addressed our initial concerns and adequately discuss remaining shortcomings in their manuscript. The authors responded to our previous concerns about the advantages and applicability of ADAPT-MS by stating that “What our study demonstrates is not incremental performance improvement over existing ML pipelines, but a fundamentally different capability: the ability to apply discovery proteomics directly to individual patient samples” and “The framework directly addresses real clinical needs: managing diagnostic uncertainty, supporting multiple differential diagnoses from a single measurement, and adapting to the pervasive issue of missing values in patient samples”. Similar to our initial review, we respectfully disagree with regard to a possible application scenario that extends from basic research to a clinical setting. We think that (a) due to the adaptive nature of ADAPT-MS, its models cannot be sufficiently validated or meet regulatory requirements, and thus are not going to be applicable in clinical use; (b) targeted or hybrid targeted-exploratory strategies can ensure that features are detected in new patient samples and make the ADAPT-MS approach redundant; (c) already the initial feature selection from reference cohorts should consider feature robustness by applying, e.g., strict sparsity reduction filtering.

We appreciate the effort of translating knowledge from proteomic cohorts directly to support clinical decision making. We appreciate the intention and novelty of ADAPT-MS and deem it publishable. Nevertheless, we stand to our initial viewpoint that it lacks wider clinical significance and hence might be more adequate for a more specialized readership. At the same time, we respectfully acknowledge that this decision is ultimately at the discretion of the Editorial Board.

(Remarks on code availability)

Reviewer #4

(Remarks to the Author)

(Remarks on code availability)

Point-by-point answers to REVIEWER COMMENTS for An adaptive, continuous-learning framework for clinical decision-making from proteome-wide biofluid data

Reviewer #1 (Remarks to the Author):

The manuscript proposes the ADAPT-MS framework for sample-specific model retraining on discovery-mode proteomics data. While the concept of avoiding fixed biomarker panels is innovative, a number of methodological choices and presentation issues must be addressed before the work is suitable for publication.

We thank the reviewer for their positive and constructive review of the paper. We believe we have addressed the remaining concerns below.

Major Comments

1. In the first “fixed feature selection” stage the authors carry out full-cohort imputation, yet this imputed data strongly influences the downstream relaxed feature list. What evidence supports applying imputation only at this stage and not during the subsequent single-sample retraining? A detailed justification—including sensitivity analyses that vary the initial imputation strategy—would strengthen the manuscript.

We thank the reviewer for highlight the important point of imputation. However, we believe the reviewer’s question may be based on a misunderstanding:

Indeed, imputation is often required as many ML algorithms required complete data matrices and ADAPT-MS does this too, but only after the feature selection. This is used for model training and performance assessment on the discovery cohort. However, to generate the relaxed feature list via a t-test, no imputation is required. Likewise, and importantly, imputation is not used in the single sample assessment – a defining property of ADAPT-MS. Instead, we select only the features (proteins) that are present in this single sample, which gets around all the intricacies and potential biases of imputation.

In the revised manuscript, we now make this clearer by adding the following text:

“This dynamic retraining strategy overcomes the long-standing problem of missing values **which are always present in any** discovery-mode proteomics. **Afterwards, when applying a model on a single sample of interest, we only rely on features (proteins) that are actually detected. In this way, ADAPT-MS sidesteps the intricacies and potential biases of imputation or a simple zero-fill approach.** It enables robust single-sample classification without requiring complete data matrices or bias-prone imputation, and allows direct diagnostic use of the proteomics measurement itself on single samples – transforming the output of a discovery experiment into a clinical decision tool.”

2. Instead of dropping unobserved proteins from the fixed feature list and retraining, why not retain the original fixed model (e.g., the XGBoost model presented in the text) and simply set absent feature values to zero? Please report the diagnostic performance of this “zero-fill” approach so that readers can judge the added value of sample-specific retraining.

In the course of developing and validating ADAPT-MS we had already tested the framework against a number of architectures. However, given the degrees of freedom and possibilities to apply within those it was not possible to do this comprehensively. Given the reviewer's excellent suggestion, we have now compared their suggested architecture against our ADAPT-MS pipeline in the figure below: Left to right: RF + XGB architecture on imputed data frames (ROCAUC: 0.83); suggested architecture with trained XGB model and zero-fill approach for missing values of validation samples (ROCAUC: 0.82); ADAPT-MS architecture with t-test feature selection and logistic regression (ROCAUC: 0.87). Thus while we agree that other architectures are certainly possible, they are not superior.

3. The authors cite algorithmic simplicity as the reason for choosing logistic regression, yet logistic regression is a linear classifier. What empirical or theoretical evidence suggests that linear decision boundaries are adequate for these proteome-wide clinical tasks? A comparison to at least one nonlinear model under identical training conditions is warranted.

We thank the reviewer for bringing up the topic of what models can or should be used in ADAPT-MS. We have indeed extensively tested non-linear models, as the reviewer suggests (see Suppl. Fig. 1). When we empirically tested combinations of different feature selection and classifier combinations, LR had the overall highest performance in combination with various feature selection methods. Given this, we prefer the simplest and presumably most robust method, which manifests in not overfitting the training datasets. This was the main rationale for choosing LR as a classifier in the paper-presented examples. That said, the ADAPT-MS framework architecture is algorithm agnostic towards the feature selection and classification. We now make this more clear in the manuscript by the following statement:

“However, the ADAPT-MS framework is agnostic to both the feature selection method and classification algorithm employed. Classification tasks that requiring non-linear decision-boundaries can instead incorporate algorithms such as random forest or gradient boosting within our ADAPT-MS architecture, simply by substituting the classifier while maintaining the sample-specific retraining strategy.”

4. The discussion highlights the ability to reuse discovery-mode data for exploratory diagnostics, but does not convincingly explain why this is preferable to more economical targeted proteomics once a clinically actionable protein set has been identified. Please clarify the cost-benefit rationale.

We appreciate the reviewer's point but we believe that we already go out of our way to contextualize the advance presented here with respect to current practice. We believe that the key advantage of ADAPT-MS is not replacing targeted assays for established biomarkers, which remain the gold

standard for specific diagnostic applications, but rather enabling perspectives that targeted analysis by its nature is incapable of:

1. Multiplexed diagnostics from a single measurement: Unlike targeted assays that answer one clinical question, ADAPT-MS can address multiple differential diagnoses sequentially without remeasurement. This is particularly valuable in complex clinical presentations where the diagnostic pathway is unclear.
2. Rapid deployment for emerging conditions: New diagnostic needs (e.g., novel disease subtypes, emerging pathogens) can be addressed immediately using existing proteome data, without the multi-year development cycle required for targeted assays.
3. Continuous improvement: As the reference database grows, all diagnostic applications improve automatically, whereas targeted assays remain static once validated.

Thus while we agree that a detailed health economics analysis comparing discovery proteomics with targeted approaches would be valuable, as an academic proof-of-concept, our primary goal is to demonstrate technical feasibility and clinical potential. The cost-effectiveness will ultimately depend on factors such as throughput, automation, and clinical implementation models that are beyond the scope of this initial study.

For the near future, we envision ADAPT-MS complementing rather than replacing targeted assays, thus, serving as a discovery and triage tool that can identify when specific targeted tests are warranted, while also addressing diagnostic questions for which no targeted assays yet exist.

In the revised manuscript we added:

“We believe the key advantage of ADAPT-MS is not to replace targeted assays for established biomarkers - which remain the gold standard for specific diagnostic applications - but to enable capabilities that targeted approaches cannot. First, unlike targeted assays that answer a single clinical question, ADAPT-MS can support multiple differential diagnoses from one measurement, which is particularly valuable when clinical presentations are complex or ambiguous. Second, it allows rapid deployment for emerging diagnostic needs (e.g., novel disease subtypes or pathogens) without the lengthy development cycle required for targeted panels. Third, as reference databases expand, ADAPT-MS applications improve continuously, whereas targeted assays remain fixed once validated.”

5. The Abstract states that the framework enables both diagnostic and prognostic interpretation, yet the manuscript offers no explicit prognostic example. Please provide at least one prognostic use-case (e.g., survival or treatment response prediction) or temper the claim.

We thank the reviewer for this comment and agree that we had indeed not provided an example for prognostic applications. Prompted by the reviewer’s comment, we now include an additional study to prove that ADAPT-MS performs well for prognostic predictive tasks, too. We have added the following paragraph to the text:

“To illustrate the extension of ADAPT-MS to prognostic classification tasks, we applied our architecture to a recently published cohort investigating the risk prediction for development of metabolic syndrome (MetS) from serum proteomics²⁹. Both discovery and validation cohorts were measured using unbiased discovery proteomics, making this dataset an ideal example case. The

authors of the study applied a complex ML architecture wrapped around LightGBM gradient boosting; thus, it is not surprising that a standard RF – XGBoost model does not perform on par (Fig. 2D). However, the ADAPT-MS architecture yields equivalent performance to the published model (AUC 0.77) on the validation data, demonstrating the power of our simple and explainable yet readily applicable model (Fig. 2E).”

6. In Fig. 2C the training-set performance of ADAPT-MS decreases as the number of training samples increases. This counter-intuitive trend needs explanation—does it reflect over-fitting at small sample sizes, class-imbalance effects, or something else?

The reviewer raises an excellent point. This trend comes from the fact that chunks of data are added consecutively to the experiment with this procedure, the training set intrinsic train-test split can be favorable or unfavorable for performance readout and therefore ‘wobble’. In bootstrapped ROC curves this is usually figured in the plot by an error bar, in this case, only one iteration of train-test split was done per sample size data point. Therefore, this is one draw from a distribution. We have now changed this and present the median of a 10x bootstrap per sample size datapoint, which is more stable towards stochastic sampling and creates smoother outcome data.

7. Line 249 claims that the method “does not rely on full-cohort imputation or harmonization,” yet the relaxed feature selection explicitly uses full-cohort imputation. This sentence should be revised to accurately describe the workflow.

We agree – please see our answer to the imputation point above. As requested by the reviewer, we have now changed the sentence as follows:

“Importantly, the method does not rely on full-cohort imputation or harmonization for any validation or actual application case on single samples, but adapts per sample based on the available feature space (Fig. 3B).”

8. ADAPT-MS performs worse within the Berlin cohort than between cohorts. Please explain this pattern (e.g., batch effects, cohort heterogeneity) and include the corresponding fixed-feature XGBoost results for parity.

We thank the reviewer for highlighting this inconsistency. The difference in the Berlin is already commented on in our original paper by Bader et al.: ‘In the Berlin cohort, proteome alterations between AD and non-AD CSF were smaller with only 168 proteins exhibiting significantly ($P < 0.05$) different abundances’ as compared to the 540 and 453 significantly changed proteins in the Sweden

and the Magdeburg/Kiel cohorts. This likely reflects its different control population (subjective cognitive impairment and depression patients), which makes within-cohort classification intrinsically harder.

The fixed-feature XGBoost approach applied within the Berlin cohort yields the expected low performance, though this can be partly improved when training on all cohort subsets. The accompanying figure (left to right) shows: training performance on a 20-sample Berlin subset, validation performance from training on the same 20-sample subset, and validation performance when training on all 20-sample subsets.

This question also came up with Reviewer 3, and for additional information in the revised manuscript, please see below.

Minor Comments

1. Line 194: “Extended Data Fig. 2E” should read “Extended Data Fig. 1E.”
2. Clearly state the training and test sample counts for the Alzheimer’s disease cohorts.
3. Lines 271–273: Add the full reference for Mi et al.

We thank the Reviewer for finding these errors. They are addressed in the revised manuscript.

Reviewer #2 (Remarks to the Author):

Reviewer #3 (Remarks to the Author):

Improved MS instrumentation has markedly increased the depth of proteomic information that can be obtained from a single sample and enables high-throughput workflows, building up on population databases. Additionally, the potential of machine learning models for the analysis of complex omics data is omnipresent. At present, many studies report on the use of machine learning models for clinical decision-making based on proteomics data.

We thank the reviewer for this thoughtful comment. We agree that recent advances in MS instrumentation and machine learning have greatly expanded the possibilities for clinical proteomics, and that many studies already report on ML applications in this field. What distinguishes ADAPT-MS is that it moves beyond models trained and validated only within carefully curated study datasets. In most existing work (including our own previous studies) ML models perform well in the context of a fixed cohort but cannot be applied reliably to a newly measured single patient sample.

ADAPT-MS addresses this gap by dynamically retraining classifiers on the proteins actually quantified in each sample, thereby enabling diagnostic use at the level of the individual patient rather than at the level of the study cohort. We agree that clinical translation remains a long-term goal, but we view this as an important first step toward a truly sample-by-sample ML framework. To clarify this distinction, we have fully revised Figure 1 to illustrate how ADAPT-MS differs from conventional biomarker pipelines and from direct application of study-intrinsic ML models.

In this manuscript, the authors present a framework for adaptive model generation. In principle, a classification model is first trained on reference data using an initial list of proteins. Before the model is applied to a single, unlabeled test sample, this list of proteins is adjusted if these proteins are missing in the test sample. The adjusted list is then used to re-train the model on the reference data before classification of the test sample.

In a first example of application, the authors demonstrate the classification of sepsis patients based on plasma proteomics data. Conventional classification models using the XGBoost algorithm and ADAPT-MS models both show good performance with AUC values in the range of 0.83-0.87. As shown in Fig. 2C, the performances of XGBoost and ADAPT-MS models slightly vary depending on the number of samples in the training set. XGBoost models perform better for training sets of 200-500 samples and ADAPT-MS performs better for training sets of 500-900 samples, though the difference in performance is small.

We agree with this succinct summary of our findings by the reviewer.

In a second example, the authors claim to showcase generalizability across three clinical centers that represent real-world variability. For the application of machine learning models in clinical proteomics, the integration of datasets from independent sites is indeed a crucial challenge. However, the data provided in Fig. 3A shows that XGBoost classification models without adaptive learning match or perform better than ADAPT-MS models in all cases. In addition, page 8, line 224 states that the independent cohorts were processed under “comparable protocols”. While this might be true in terms of sample collection at the different clinical sites, all samples were processed in the same laboratory using identical protocols. In the original publication by Bader et. al, the Methods sections provides information that all cohort samples were processed together and MS data was acquired on the same instrument using the same settings. Yet, most variability in proteomic data results from different sample preparation protocols and different MS instrumentation. In my opinion, the data presented in the manuscript therefore does not demonstrate “real-world variability” in proteomics data of independent sites. The application of a classification model, which was trained on a reference dataset, and tested on a single, independent sample of different origin, is still questionable. I would encourage the authors to demonstrate the application of the developed software on datasets that were actually processed independently in different clinical sites.

We thank the reviewer for this critical evaluation. Indeed, the three cohorts of the AD study were processed under comparable protocols in terms of preprocessing as well as sample preparation and MS measurements. However, a complication of our study was that we measured the Sweden cohort three months after the other ones and on a different MS instrument (this is also apparent from the publicly available raw data files). Conversely, such complications are not uncommon in proteomics practice and they give us a proving ground for the robustness of ADAPT-MS.

The Sweden vs other ‘batch effect’ separation is visible in PCA space, where Sweden samples cluster distinctly along PC1, indicating a processing or instrument-related batch effect (see figure below). Importantly though, the Sweden and Magdeburg/Kiel cohorts show similar AD–non-AD separation, whereas the Berlin cohort displays weaker biological contrast, consistent with its different control population (Bader et al. Figure 3c-e).

In summary, these three cohorts together form a semi-overlapping test space: one intrinsically different (Berlin), one processed and measured separately (Sweden), and one intermediate (Magdeburg/Kiel). This provides a useful, if imperfect, proof-of-concept for ADAPT-MS generalization.

We agree with the reviewer that a true inter-center dataset would be the best test for the principles of ADAPT-MS. Such datasets, however, remain extremely rare because external validation cohorts are seldom acquired with comparable discovery proteomics workflows. We have added the PCA plot as well as the following statement to the text:

“The variability is visible by cohort-specific proteome signatures in unsupervised methods (Extended Data Fig. 2J) as well as in the different numbers of significantly changing proteins between AD patients and control samples across the different cohorts.”

In summary, the authors provide a concept on how to enable fast use of proteomics discovery cohorts without the need for time-consuming analysis of validation cohorts. As the authors describe in the Discussion, implementation for clinical decision-making requires robust and reproducible validation. However, adaptive learning is quite the opposite and will hardly meet regulatory requirements.

We thank the reviewer for raising this critical point. We agree that ADAPT-MS is at an early stage and that strict validation against regulatory standards will be essential before clinical use.

We now include this clarifying statement in the Discussion:

“However, it is clear that for implementation in clinical practice, this principle will have to be applied to real-world validation studies before our presented vision of the tailored single sample use-case can be approved and readily applied.”

Further, the data provided in this manuscript does not sufficiently demonstrate the advantage of adaptive classification models over conventional, previously published approaches. Especially in the context of clinical decision-making that requires strict, yet important regulations.

Further, the strategy of adjusting feature lists and re-training models provides little novelty compared to ever-advancing machine learning approaches in proteomics data analysis.

The comment of the reviewer is important. What our study demonstrates is not incremental performance improvement over existing ML pipelines, but a fundamentally different capability: the ability to apply discovery proteomics directly to individual patient samples. Conventional models are trained and tested within closed, self-sustained datasets; they cannot be prospectively applied to a single new sample without imputation or panel reduction. ADAPT-MS overcomes this barrier by retraining per sample on measured proteins, thereby making discovery proteomics usable for diagnostics in principle for the first time. This is the critical advantage, and it directly addresses the gap between research proteomics and clinical application.

We add the following clarifying limitation statement in the Discussion:

“Importantly, we do not claim that ADAPT-MS outperforms other ML techniques in the place where they are usually used today, such as on self-sustained dataset in a rather analytical then diagnostic way. Rather, its novelty lies in enabling single-sample diagnostic application of discovery proteomics data — a use case for which no other framework currently exists.”

In addition, the comparison of computation expense to existing algorithms has not been shown.

We thank the reviewer for this suggestion. We have now added a direct comparison of computational expenses between conventional XGBoost and ADAPT-MS on the sepsis dataset (Extended Data Fig. 1G–H).

As shown, ADAPT-MS scales slightly less efficiently during training due to repeated feature-selection steps, but remains computationally lightweight overall and faster than XGBoost during validation (a one-time computational cost that takes only minutes on a standard laptop).

We added this statement in the text:

“Computational time increased with training set size for both methods (Extended Data Fig. 1G–H). ADAPT-MS showed somewhat steeper growth during training, reflecting repeated feature-selection steps, but remained computationally lightweight overall (a one-time computational cost that takes only minutes on a standard laptop) and was faster than XGBoost during validation.”

The concept of the presented classification algorithm is intriguing and the algorithm certainly deserves publication. However, given the aforementioned constraints, it might be adequate to rather focus on the bioinformatic advancement rather than a very speculative clinical application. A more specialized journal might be a more adequate platform.

We thank the reviewer for acknowledging that our algorithm deserves publication. We respectfully disagree that the clinical applications are speculative or that a specialized bioinformatics journal would be the better venue. Framing ADAPT-MS only as a computational advance would miss its broader significance. The framework directly addresses real clinical needs: managing diagnostic uncertainty, supporting multiple differential diagnoses from a single measurement, and adapting to the pervasive issue of missing values in patient samples. We believe Nature Communications is an ideal platform because these advances speak to a broad biomedical audience and highlight a path from research proteomics toward eventual clinical use—an audience unlikely to be reached through a purely bioinformatics outlet. We emphasize this in the manuscript:

“Looking ahead, the potential of this approach is considerable. As longitudinal, high-throughput proteomics becomes feasible at population scale and decreasing cost, individual-level proteome profiles may become part of routine clinical records. In that context, ADAPT-MS could enable real-time diagnostic interpretation by matching each new patient to precisely defined reference cohorts and applying flexible, task-specific models. This shift from assay-centric to data-centric diagnostics would align proteomics more closely with the model already established in clinical genomics and could support a new level of precision in medical decision-making.”

Further notes:

- Author affiliation 5 is not mentioned for an author
- p. 7, l. 194 “Extended Data Fig. 2E” must be replaced with “Extended Data Fig. 1E”

We thank the Reviewer for finding these errors. They are addressed in the revised manuscript.

Reviewer #4 (Remarks to the Author):

REVIEWER COMMENTS – point-by-point answers

Reviewer #1 (Remarks to the Author):

With all previous issues having been satisfactorily addressed by the authors, the manuscript is recommended for acceptance. One minor correction is requested - the upper panel of Figure 3 should be removed.

We thank the reviewer for their constructive feedback on our manuscript throughout the revision process. The duplicated figure 3 has been removed from the manuscript.

Reviewer #3 (Remarks to the Author):

The authors have mostly addressed our initial concerns and adequately discuss remaining shortcomings in their manuscript. The authors responded to our previous concerns about the advantages and applicability of ADAPT-MS by stating that “What our study demonstrates is not incremental performance improvement over existing ML pipelines, but a fundamentally different capability: the ability to apply discovery proteomics directly to individual patient samples” and “The framework directly addresses real clinical needs: managing diagnostic uncertainty, supporting multiple differential diagnoses from a single measurement, and adapting to the pervasive issue of missing values in patient samples”. Similar to our initial review, we respectfully disagree with regard to a possible application scenario that extends from basic research to a clinical setting. We think that (a) due to the adaptive nature of ADAPT-MS, its models cannot be sufficiently validated or meet regulatory requirements, and thus are not going to be applicable in clinical use; (b) targeted or hybrid targeted-exploratory strategies can ensure that features are detected in new patient samples and make the ADAPT-MS approach redundant; (c) already the initial feature selection from reference cohorts should consider feature robustness by applying, e.g., strict sparsity reduction filtering.

We thank the reviewer for their constructive feedback and critical contrasting opinion. As stated in our manuscript we acknowledge that ADAPT-MS is not a technique to replace targeted assays (‘We believe the key advantage of ADAPT-MS is not to replace targeted assays for established biomarkers - which remain the gold standard for specific diagnostic applications - but to enable capabilities that targeted approaches cannot.’). Thus, it is our aim to explore ways to perform diagnostic procedures with discovery proteomics data itself and also discuss the regulatory difficulties to still overcome. ADAPT-MS in our hands proved as a first of its kind architecture for machine learning to bring classification of patients to discovery proteomics data. Further developments will show how it can be applied in actual clinical practice.

We appreciate the effort of translating knowledge from proteomic cohorts directly to support clinical decision making. We appreciate the intention and novelty of ADAPT-MS and deem it publishable. Nevertheless, we stand to our initial viewpoint that it lacks wider clinical significance and hence might be more adequate for a more specialized readership. At the same time, we respectfully acknowledge that this decision is ultimately at the discretion of the Editorial Board.

We thank the reviewer for their feedback and agree that research results and the relevance to different fields is difficult to foresee. We aim to build on this groundwork and hope to convince the reviewer with the future publication of actionable and practicable examples of the clinical significance of our work.

Reviewer #4 (Remarks to the Author):

We thank the reviewer for their time and effort spent on reviewing our manuscript.